# Acute and Chronic Effects of Fin Amputation on Behavior Performance of Adult Zebrafish in 3D Locomotion Test Assessed with Fractal Dimension and Entropy Analyses and Their Relationship to Fin Regeneration

**DOI:** 10.3390/biology11070969

**Published:** 2022-06-27

**Authors:** Gilbert Audira, Michael Edbert Suryanto, Kelvin H.-C. Chen, Ross D. Vasquez, Marri Jmelou M. Roldan, Chun-Chuen Yang, Chung-Der Hsiao, Jong-Chin Huang

**Affiliations:** 1Department of Chemistry, Chung Yuan Christian University, Chung-Li 320314, Taiwan; gilbertaudira@yahoo.com (G.A.); michael.edbert93@gmail.com (M.E.S.); 2Department of Bioscience Technology, Chung Yuan Christian University, Chung-Li 320314, Taiwan; 3Department of Applied Chemistry, National Pingtung University, Pingtung 900391, Taiwan; kelvin@mail.nptu.edu.tw; 4Department of Pharmacy, Research Center for Natural and Applied Sciences, University of Santo Tomas, Manila 1008, Philippines; rdvasquez@ust.edu.ph; 5Faculty of Pharmacy, The Graduate School, University of Santo Tomas, Manila 1008, Philippines; mmroldan@ust.edu.ph; 6Department of Physics, Chung Yuan Christian University, Chung-Li 320314, Taiwan; chunchuenyang@cycu.edu.tw; 7Department of Physics, National Central University, Chung-Li 32001, Taiwan; 8Center of Nanotechnology, Chung Yuan Christian University, Chung-Li 320314, Taiwan; 9Research Center for Aquatic Toxicology and Pharmacology, Chung Yuan Christian University, Chung-Li 320314, Taiwan

**Keywords:** zebrafish, fin amputation, behavior, pain, lidocaine

## Abstract

**Simple Summary:**

Fin amputation is a routinely conducted procedure for various experiments, especially in zebrafish. However, no study compares the acute and chronic effects of the amputation of each fin on their behaviors. In addition, although some analgesics have been applied after the fin amputation procedure, the long-term effects of these drugs in have not been evaluated yet. In this study, we found that amputation in the caudal fin resulted in the most pronounced behavior alterations and their behavior was fully recovered before the caudal fin was fully regenerated, indicating that these behavioral changes came from pain elicited from the fin amputation. Finally, while lidocaine treatment could ameliorate the behavioral effects after the amputation procedure, it did not accelerate the behavior recovery process; instead, it caused the fish to display some slight side effects.

**Abstract:**

The fin is known to play an important role in swimming for many adult fish, including zebrafish. Zebrafish fins consist of paired pectoral and pelvic with unpaired dorsal, anal, and caudal tail fins with specific functions in fish locomotion. However, there was no study comparing the behavior effects caused by the absence of each fin. We amputated each fin of zebrafish and evaluated their behavior performance in the 3D locomotion test using fractal dimension and entropy analyses. Afterward, the behavior recovery after the tail fin amputation was also evaluated, together with the fin regeneration process to study their relationship. Finally, we conducted a further study to confirm whether the observed behavior alterations were from pain elicited by fin amputation procedure or not by using lidocaine, a pain-relieving drug. Amputation in the caudal fin resulted in the most pronounced behavior alterations, especially in their movement complexity. Furthermore, we also found that their behavior was fully recovered before the caudal fin was fully regenerated, indicating that these behavioral changes were not majorly due to a mechanical change in tail length; instead, they may come from pain elicited from the fin amputation, since treatment with lidocaine could ameliorate the behavioral effects after the amputation procedure. However, lidocaine did not accelerate the behavior recovery process; instead, it caused the fishes to display some slight side effects. This study highlights the potential moderate severity of fin amputation in zebrafish and the importance of analgesia usage. However, side effects may occur and need to be considered since fin amputation is routinely conducted for various research, especially genomic screening.

## 1. Introduction

Fish need to coordinate movements across their body in capturing prey, avoiding predators, and navigating their environment. This coordination involves fluid movements of the jaws, eyes, tail, and fins [1]. There are a wide diversity of locomotor behaviors of fish, such as rapid C-start escape response, steady undulatory locomotion, and linear accelerations; however, generally, it is subdivided into swimming with paired or median fins or with the body axis and caudal fin [2,3]. In zebrafish, paired fins include the pectoral and pelvic fins, which are homologs of forelimbs and hindlimbs, respectively, of land-dwelling, limbed vertebrates [3]. In contrast, the dorsal, caudal, and anal fins, internal support elements (radial), and external supports (fin rays) belong to median fins [4,5]. The pectoral fins of juveniles and adults of many fish species act as primary propulsors during rhythmic swimming and in arrhythmic movements such as braking and maneuvering [6]. They also have been shown to undergo various motions, depending on the species and the type of movement being executed [1,7]. Generally, the pectoral fins actuate in-phase (synchronously) slow swimming [8]. Meanwhile, intermittent caudal fins assist the pectoral fin locomotion to achieve higher relative speeds [9]. Tucked pectoral fins along the sides of the body and axial movement alone can generate propulsive thrust [10]. In addition, the caudal fin is often observed to sustain damage because of predation, disease, or social rank [11]. In adult zebrafish, as in many other species, axial body bending, together with the fins, is used at all speeds to maneuver and stabilize [10]. Therefore, damage to fish’s fins might affect their locomotor performance and compromise their survivability.

Despite the importance of fins in fish, unfortunately, very few studies have evaluated in detail the effect of fins amputation on fish locomotor activity, especially zebrafish. This might be because, in most of these studies, the interpretation of the results is very challenging since the power of the fish to compensate for such deficiencies after amputations by suitable action of the remaining fins on the general equilibrium is so great [12]. Moreover, most of the studies mainly focused only on a single or several fins. Nevertheless, several studies have built a foundation to highlight the significance of fins in fish locomotion. For example, using sockeye salmon (*Oncorhynchus nerka*) as an animal model, a previous experiment examined drag and thrust energetic relations for a swimming fish after caudal fin amputation. From the results, a significant reduction in the critical speed of fish was observed when both epaxial and hypaxial caudal fin lobes were removed [13]. Furthermore, a prior study compared the swimming performances of eight groups of rainbow trout (*Salmo gairdneri*) with various fin rays removed. Later, significant differences in the performances between every group were observed, confirming the hydromechanical theory that large fins are required for acceleration [14]. In addition, research on American dogfish (*Mustelus canis*) demonstrated the instability of the equilibrium in the horizontal and vertical planes without fins by using the wind tunnel. Meanwhile, in the vertical plane, the instability is significantly increased by the presence of the pectorals, indicating the significance of fins for the fish to achieve the normal equilibrium of swimming [12]. In zebrafish, even though without fin amputation processes, the swimming performance, swimming behavior, and typical locomotor activity patterns in three of the major morphotypes—wild-type, long-ginned, and no-tail—were assessed by using a modified Brett-type water tunnel. From the results, significantly higher activity levels were displayed by wild-type fish than those of long-finned fish, and the levels of both were significantly higher than those of no-tail fish [15]. Meanwhile, malformations, including malformed fins, were also found to affect the locomotor efficiency of zebrafish larvae. The locomotor activity of the deformed larvae was markedly decreased in both the light and dark phases of testing [16]. However, to the best of our knowledge, there has been no study comprehensively comparing the differences between each amputated fin in altering the locomotion of adult zebrafish.

In recent years, video-based automatic methods have been broadly applied to monitor the behaviors of aquatic animals, focusing on swimming mechanics and the detection of multiple subjects in shoaling studies [17]. These methods actively replace manual observation due to their high overall validity, objectivity, and consistency of collected data, and the number of assessed behavioral endpoints [18,19,20,21]. Furthermore, the characteristics of these methods (e.g., high sampling frequency and high spatial resolution) make them suitable for extracting neurobehavioral phenotypes of zebrafish since, unlike rodents and humans, the locomotion of zebrafish and many other aquatic animals occurs in 3D (X, Y, Z) coordinates, generating complex behaviors and multidimensional datasets [17,20,21,22,23]. Since their biological activities are complex and inconsistent, conventional methods such as Euclidian geometry are insufficient to describe them [24,25]. Therefore, together with entropy analysis, complex geometry such as fractal geometry is required [26].

Fractal dimensions (FD) and entropies are usually utilized to measure complex systems in the real world. Fractal dimensions characterize fractal patterns by featuring the complexity as a ratio of the change in detail to the change proportionally. It is commonly applied to determine the self-similarity properties and natural structural patterns [27]. Lately, this measurement has been used to describe behavioral patterns of various animal models, including fruit flies and chironomids [28,29]. In zebrafish, FD reduces the 3D complex swimming trajectories of zebrafish to one value and was used in a previous study to quantify their swimming behavior provoked by the presence of hypochlorite in the water by using FD analysis [26]. It has also been applied to various painful treatments in zebrafish to produce an arbitrary pain intensity scale [30,31]. However, even though FD is used for behavioral monitoring, it mainly presents the overall trend of behavioral changes that are limited in presenting global and local information of movement changes after exposure to stimuli. Therefore, entropy, which is also a nonlinear measurement to describe the uncertainty and degree of disorder, is used to help the researchers in observing the behavior changes from different perspectives since it is useful for detecting changes in behavioral states overall and in specific changes including the transient periods under stressful conditions [32]. In addition, it also has been applied in complex biological systems [33]. Understanding the original meaning of entropy is crucial to elucidating how entropy applies in biology. Bandt and Pompe first proposed the concept of permutation entropy, which considers the data sequence [34]. Thus, the obtained entropies are highly correlative to the data point next nearest neighbor. They also recommended calculating the permutation entropy in the relative lower embedding dimensions, the low-dimensional space that translates from high-dimensional vectors. These basic assumptions reduced the computational complexity and increased the readability of the information. Since it measures the predictability of the value of a variable, in our case, the higher the difficulty in predicting the fish position means the higher the entropy [35]. Permutation entropy has been proven helpful in a wide range of applications where measurement of complex time series where needed [33]. For example, in a prior publication, it was applied to analyze the behavioral changes of the fruit fly in response to stressors [32]. We believe that combining these approaches can give different perspectives from the commonly used behavior endpoints in observing zebrafish behavior alterations after treatments.

Addressing the significance of the fins of fish, we investigate the effects of several fins’ amputation on fish locomotion by combining a 3D locomotion video-based automatic method, fractal dimensional, and entropy calculation from the continuous data to reflect changes in movement behaviors. Using this combined approach, this paper is the first to study detailed three-dimensional movements of fin-amputated fish and evaluate hypotheses for how these fins might function. We hypothesized that each fin affects the stability of the equilibrium either in the horizontal or vertical planes of zebrafish swimming. Based on those data, we found that the absence of one type of fins caused more severe behavior alterations in the fishes than other fins. Later, we conducted a more profound investigation regarding their recovery process after amputating this fin in terms of their behaviors in the 3D locomotion test. In addition, the potential cause of the behavior alterations was also studied by using lidocaine, a local anesthetic with analgesic properties. The overview of the current study is shown in Figure 1.

## 2. Materials and Methods

### 2.1. Zebrafish Maintenance

In this study, adult zebrafish (4–6 months old) of two strains, which were pet-store-purchased (PET) and AB, were used. The AB strain, which was originated from Taiwan Zebrafish Core Facility at Academia Sinica (https://sl.icob.sinica.edu.tw/tzcas/ (accessed on 12 June 2022)), had been reared in the zebrafish facility for more than twenty generations, while PET zebrafish were obtained from the local aquarium store and reared in the zebrafish facility for at least one month prior to the experiment. Even though readily acclimatizing to the new environment is one of the zebrafish traits, this process is still recommended for the PET zebrafish to eliminate the external factors, including the stressful condition during the transfer [36]. Zebrafish were kept in a recirculating aquatic system at 28 ± 1 °C with a 10/14 h dark/light cycle and pH 7.0–7.5. The conductivity of the water was maintained between 300 and 1500 µS, and it was constantly filtered by ultraviolet (UV) light. Every day, the fish were fed twice with lab-grown brine shrimp and commercial dry food, each once a day. The routine housing conditions and maintenance procedures were based on the published protocol [37].

### 2.2. Fin amputation and Treatments

Zebrafish of mixed gender were used in the current study. The study was divided into three sections that started with the behavior evaluation after the amputation of each fin. In this section, a total of ~130 PET zebrafish with similar body sizes and shapes were selected from the holding tanks using a simple random allocation method to avoid experimental bias [38]. PET zebrafish were used in the first section of this study to reduce the usage of AB zebrafish, considering that there is a limitation in the number of AB zebrafish [39]. In addition, the current PET zebrafish also possess a similar size and morphology to AB zebrafish and have been used in various behavior studies [40,41]. Later, the fish were divided into six groups, which were the control group (all fins intact), dorsal fin treatment group (dorsal fin amputation), caudal fins treatment group (caudal fins amputation), anal fin treatment group (anal fin amputation), pelvic fin treatment group (pelvic fin amputation), and pectoral fin treatment group (pectoral fin amputation), with ~21 fishes for each group. Before the amputation, each fish was transferred to a beaker glass with 500 mL of clean and dechlorinated water and was anesthetized with 0.1% MS222 tricaine (A5040, Sigma, St. Louis, MO, USA), which was determined by a loss of righting reflex (at least 30 s). Afterward, the fins in each treatment group were transversally amputated with sharp, sterile scissors [13]. A single drop of amoxicillin in 0.5 ppm of concentration was applied at the amputation site before the fish was returned to the holding tanks for ~48 h to recover before any experimental measurement. The whole amputation process was based on the previous protocol [11]. The experiment was performed in triplicates with a total *n* number of ~21 fish for each group. This sample size for each was chosen based on the statistical power analysis results that were calculated by using $n={\left(\frac{Z\sigma }{E}\right)}^{2}$ in the untreated group with a 95% confidence interval (CI) and a margin of error of 7 ± 1 units, showing that a sample of size ~17 was needed [42]. In addition, Cohen’s *d* method was used to calculate the size of the difference between two groups and found that the average effect size values in both group and individual data were 3.23 and 0.63, respectively [43]. Based on the results of this section, the absence of a caudal fin caused the most robust behavior alteration in the zebrafish; therefore, we proceed to the next section of the experiment by using the caudal fin as the main interest. This section used AB zebrafish since this strain is a widely used laboratory WT strain. A total of ~36 fishes were randomly selected from the holding tanks and divided into two groups: the control group and the caudal fins treatment group. The amputation procedure was according to the protocol in the previous section. The experiment was performed in triplicates with a total *n* number of ~18 fish for each group. Next, a new batch of AB fish was prepared to confirm whether the behavior alteration was potentially caused by the pain elicited from the fin amputation procedure. This batch was randomly assigned to one of the two treatment groups: a rescue group subject to caudal fin amputation administered with lidocaine (5 mg/L dissolved in the tank water, L7757 Sigma Aldrich, St. Louis, MO, USA) and a group that was only treated with lidocaine in the same concentration with a total *n* number of ~18 fish for each group. The concentration of lidocaine was chosen based on previous publications [30,44,45]. The experiment was also conducted in triplicates. All experimental procedures and protocols were authorized by the Committee for Animal Experimentation of Chung Yuan Christian University Committee (CYCU110016, 29 December 2021). All animal tests were carried out following the regulations released by CYCU’s Institutional Animal Care and Use Committees (IACUCs).

### 2.3. Three-Dimensional Locomotion Test and Locomotion Trajectories Analysis

The locomotion test was performed between 10:00 to 16:00 at a water temperature of 28 ± 1 °C in a quiet, temperature-controlled room (25 ± 1 °C). For the first section of the study, after ~48 h of recovery, the 3D locomotion test was conducted for the grouped zebrafish with a shoal size of 6 fishes. Afterward, the experiment was continued with the individual zebrafish 2–4 days after the grouped zebrafish test. For this test, a polypropylene box (20 × 20 × 20 cm) was used as the test tank with light-emitting diode (LED) light on the bottom and the back of the tank as the background light sources. After the test fish was put inside the test tank, it was allowed to acclimate for ~10 min before the video recording started. A Canon EOS D600 digital single-lens reflex camera with a 55–250 mm lens (Canon Inc., Tokyo, Japan) was placed ~6 m in from of the test tank and used during the video recording process. Later, after the fish behaviors were recorded for 5 min, the videos were transferred to a computer desktop. The resolution of the video was 1280 × 720 pixels at 50 frames per second (fps). Subsequently, idTracker (http://www.idtracker.es/), a movement tracking software, was used to analyze zebrafish locomotion trajectories during the test [46]. Finally, several important behavior endpoints were calculated based on the trajectories. The 3D locomotion test used in the present study was according to a previously described protocol and the description of each behavior endpoint was listed in Table A1 [47]. Next, since we did not find any significant statistical difference in the individual test results from the first section, we decided to conduct only a grouped test in the following section. In this section, the behaviors of the fishes were evaluated by the 3D locomotion test on 1 h, 3 h, 1 day, 2 days, 3 days, 5 days, 8 days, 10 days, 15 days, 20 days, 25 days, and 30 days post-amputation (dpa) to observe the recovery process of the amputated fishes in terms of their locomotion. In addition, based on a previous experiment, these time points were applied since a prior study demonstrated that shorter measuring intervals and observation periods significantly affect opercular movements, which may affect data validity [48,49]. The immediate responses observation was performed since a previous study found that tricaine does not affect several commonly used behavioral parameters, indicating the unnecessary postponing of behavioral observations to 30 min after anesthesia [50]. The 3D locomotion assay was carried out according to the protocol mentioned above and the major behavioral endpoints and their definition were summarized in Table A1. In addition, the regeneration process of the caudal fin was also monitored by capturing the image of the fin every 5 days. Prior to the process, fishes were anesthetized with 0.1% MS222 tricaine and placed into a Petri dish with a molded agarose to maintain their position. Afterward, images of the fishes were captured, and the fishes were quickly transferred to a tank with fresh water for recovery. The captured images were quantified by measuring the area of the fins in comparison with the area of the fins at day 30 since the regenerated fin should have regrown to its original length during that time [51].

### 2.4. The Mathematic Calculation for Fractal Dimension

The correlation dimension is used in this paper since fish swim in the water and change their direction at every moment. The raw data contain the time record and the three-dimensional positions of *x*, *y*, and *z*, each point following one another. The *i*th of the distance of consecutive points is defined as
$$d_{i}=\sqrt{{\left(x_{i+1}-x_{i}\right)}^{2}+{\left(y_{i+1}-y_{i}\right)}^{2}+{\left(z_{i+1}-z_{i}\right)}^{2}}$$

Here, we can define *r*, N_1_(*r*), and N(*r*), representing “any real number”, the numbers of *d_i_* that are larger than *r*, and the total numbers of *d_i_*.

The correlation function C(*r*) is defined as

C(*r*) = N_1_(*r*)/N(*r*).


If *r* is too large, then N_1_(*r*) will equal N(*r*), and logC(*r*) becomes zero. Otherwise, if *r* is too small, then C(*r*) is zero, which cannot represent the system’s status. The proper *r* value is critical in the calculation.

The fractional dimension *D* is defined as
$$D=\underset{x\to 0}{\lim}\frac{\log C\left(r\right)}{\log\left(r\right)}$$

We can find the *D* value at the logC(*r*) to log(*r*) plot near *r* = 0 using linear regression. The slope is the correlation fraction dimension, *D*.

### 2.5. The Mathematic Calculation for Entropy

The meandering entropies are calculated from the position vector pairs. The position vector of consecutive points is defined as Δ*r* = (*x_i+_*_1_
*− x_i_*, *y_i+_*_1_
*− y_i_*, *z_i+_*_1_
*− z_i_*). To check whether the direction changes, the law of cosines is utilized to extract the angle between two adjacent position vector pairs.
$$\theta =\cos^{-1}\frac{\left(x_{i}-x_{i-1}\right)\left(x_{i+1}-x_{i}\right)+\left(y_{i}-y_{i-1}\right)\left(y_{i+1}-y_{i}\right)+\left(z_{i}-z_{i-1}\right)\left(z_{i+1}-z_{i}\right)}{\sqrt{{\left(x_{i}-x_{i-1}\right)}^{2}+{\left(y_{i}-y_{i-1}\right)}^{2}+{\left(z_{i}-z_{i-1}\right)}^{2}}\sqrt{{\left(x_{i+1}-x_{i}\right)}^{2}+{\left(y_{i+1}-y_{i}\right)}^{2}+{\left(z_{i+1}-z_{i}\right)}^{2}}}$$

Hereafter, count the number of the obtained angles that are larger than 90 degrees by using the following equation:$$H(2)=-\frac{M_{1}}{M}\log_{2}\left(\frac{M_{1}}{M}\right)-\frac{M_{2}}{M}\log_{2}\left(\frac{M_{2}}{M}\right)$$

To get the meandering entropy, here *M*, *M*_1_, and *M*_2_ are the total numbers of the angles larger than 90° and smaller than 90°, respectively. Due to the identity is only to check whether the direction is changing or not, this is the entropy of order *n* = 2 system (*H*(2)).

### 2.6. Statistical Analyses

The statistical analyses and graph plotting were conducted with GraphPad Prism (GraphPad Software version 8 Inc., La Jolla, CA, USA). Before calculating the statistical differences between control and treated groups, normality tests were carried out to test the normality of data distribution. Later, most of the normality tests indicated that the data distribution was not normal. Therefore, for the first section of the study, Kruskal–Wallis followed with Dunn’s multiple comparisons test, a nonparametric test, was used since this analysis does not require a normal distribution assumption, and the data were affected by one factor, which was the amputation treatment [52]. Meanwhile, for the results from the last two sections, mixed-model two-way ANOVA continued with Sidak’s multiple comparisons test was applied to determine how the behaviors are affected by two factors, which were the amputation treatment and time. All of the behavior endpoints data are expressed either as box and whiskers or median with an interquartile range since generally, they are used to describe data in a skewed distribution, including behavior data [53,54]. The statistic differences between the control and treated groups are indicated either with “*” (*p* < 0.05), “**” (*p* < 0.01), “***” (*p* < 0.001), or “****” (*p* < 0.0001). All of the analyses were conducted by blind-trained analysts.

### 2.7. Principal Components Analysis (PCA) and Clustering Analysis

All of the behavior endpoint values of every group were summarized using Microsoft Excel as a comma-separated values file (.csv).

Afterward, the file was uploaded to ClustVis (https://biit.cs.ut.ee/clustvis/), and to treat each variable equally, data transformation by ln(x + 1) was applied followed by unit variance scaling for each row. Singular value decomposition (SVD) with imputation method, which performs imputation and SVD iteratively until estimates of missing values converge, was used to calculate the PCA since there were no missing values on the dataset. Meanwhile, for the heatmap, Clustvis plotted using pheatmap R package, which uses popular clustering distances and methods implemented in dist and hclust functions in R, with slight modifications. The clustering started with calculating all pairwise distances and merged objects with the smallest distance in each step. Here, correlation (defined additionally as correlation subtracted from (1) was chosen as the clustering distances method, while for the linkage method, the average distance of all possible pairs was chosen for both rows and columns [55]. These methods were based on several previous behavior studies [56,57,58]. Finally, the PCA and heatmap results were saved in the computer system.

## 3. Results

### 3.1. Behavior Performance of Grouped Fin-Amputated Fishes in 3D Locomotor Activity Test

A three-dimensional (3D) locomotor activity test is a sensitive tool to assess the alterations in zebrafish swimming behavior after treatments [47,59]. Here, the swimming behaviors of the fin-amputated fishes in a group were observed 2 days post-amputation (dpa). From the results, amputations of fins affected the swimming behaviors of zebrafish. In terms of locomotion, the absence of all of the fins reduced zebrafish locomotion. This phenomenon was shown by the statistical reduction in average speed and rapid movement ratio in all treated groups (Figure 2A,D). Moreover, in most groups, the alterations in the locomotor activity were also displayed by incrementing the freezing time movement ratio (Figure 2B). Furthermore, in dorsal- and anal-fin-amputation groups, the hypoactivity was compensated by a slightly higher swimming time movement ratio than in the control group (Figure 2C). Meanwhile, regarding their movement orientation, even though none of the amputated groups showed a statistical difference in the average angular velocity, except for the caudal-fin-amputated group, the amputations affected the turning movement of the zebrafish (Figure 2E). These movement irregularities were shown by a higher level of meandering than in the control group, especially for the anal-, pelvic-, and pectoral-fin-amputated groups (Figure 2F). This phenomenon indicates that these groups often displayed zig-zag-like movements during the test. Next, similar to the locomotion results, amputations of fins led the fish to exhibit abnormal exploratory behaviors in different manners. The absence of caudal and pelvic fins caused the fishes to maintain their position at the bottom of the test tank (Figure 2H–L). In addition, loss of the several fins, including dorsal, anal, and pectoral fins, induced them to keep a distance from the center of the tank, which can indicate an anxiety-like behavior (Figure 2G).

### 3.2. Behavior Performance of Individual Fin-Amputated Fish in 3D Locomotor Activity Test

To further explore the effects of fin amputations on zebrafish swimming pattern, the zebrafish swimming behaviors in the 3D locomotor activity test was also observed individually since zebrafish possess different behaviors in the presence of conspecifics [60]. As we expected, the individual test results differed from the group test ones. Interestingly, amputations in all fins did not alter their swimming pattern, especially in locomotor activity and exploratory behaviors. This phenomenon was indicated by no statistical differences observed between treated and control groups in all locomotor activity and exploratory behavior endpoints (Figure A1A–D,G–L). However, one of the treated groups still displayed abnormal movement orientations, the dorsal-fin-amputated group. This alteration was shown by the increased level of average angular velocity in this group, even though their meandering was at a similar level to the control group (Figure A1E,F).

### 3.3. Fractal Dimension, Entropy, PCA, and Hierarchical Clustering Analyses of Grouped Zebrafish Behavior Performance after Fins Amputation

Fractal analysis was applied to measure biological structures or systems’ temporal and spatial complexity [30,61,62], while entropy was used to measure the mutual information flow between two dynamical systems [63,64]. The results showed a statistical decrement in the fractal dimension between control and several amputated groups, including caudal, pelvic, and pectoral fins in grouped zebrafish test, while this phenomenon was not displayed in the individual test ( Figure 3A and Figure A1M). Similar to this result, a statistical difference was not found in the entropy level between the control and all amputated groups while the fishes were individually tested (Figure A1N). Meanwhile, the caudal-fin-amputated group showed a statistically higher level of entropy than the control group, demonstrating the effect of this fin amputation on the zebrafish response (Figure 3B). PCA and heatmap analyses were also carried out to reduce the data dimension and complexity, thus, allowing us to explore the behavioral phenomics between every group. From both analyses, all experimental groups were divided into two major clusters. In the first cluster, anal- and dorsal-fin-amputated groups were found to belong in this cluster, including the untreated group, indicating that these two amputated groups exhibited more similar behaviors to the control group than other treated groups, which were pectoral-, pelvic-, and caudal-fin-amputated groups, which left these other treated groups to the second cluster (Figure 3C,D). In conclusion, based on these results, the caudal fin amputation had the most significant effects on the zebrafish behavior performance in the 3D locomotion test, even at 2 dpa. Therefore, it is intriguing to conduct a more profound study regarding the relationship between their behavior recovery after their caudal fin is amputated and their caudal fin regeneration.

### 3.4. Recovery of Behavior Performance of Fish after Caudal Fin Amputation in a Grouped 3D Locomotor Activity Test

Since the absence of a caudal fin was found to affect the zebrafish behaviors more severely than other fins even after ~48 h of recovery, it is intriguing to understand how early their behaviors are recovered after the amputation. Therefore, a periodic observation was conducted on the amputated zebrafish by the grouped 3D locomotor activity test. From the results, in terms of locomotion, the behavior recovery was approximately started at 1 dpa, with almost complete recovery on day 5 and full recovery on day 10 (Appendix A). This assumption was taken based on average speed, freezing, swimming, and rapid movement time ratios endpoints that showed a statistically similar level of these behavior endpoints from day 10 onwards (Figure 4A–D). A similar phenomenon also occurred regarding their movement orientation. Based on the results, the amputated fishes swam in a similar movement orientation with the untreated fish at 5 dpa, indicated by similar levels of average angular velocity and meandering started on day 5 (Figure 4E,F). Interestingly, a similar recovery time was observed in several measured exploratory behavior-related endpoints, including average thigmotaxis, total distance traveled in the top, and the number of entries to the top (Figure 4G–I). Meanwhile, overall, regarding the time spent in the three compartments of the test tanks, which were top, middle, and bottom, we did not find any statistical differences during the 30 days observation, which was different to the results from the first section (Figure 4J–L). Similar to the behavioral endpoint results, a comparable gradual recovery curve was also observed in the FD and entropy results. Here, the amputated fishes displayed a similar FD and entropy values at 3 dpa (Figure 4M,N). To sum up, the behavior performances of zebrafish in the 3D locomotion assay had almost fully recovered in 5 days after the caudal fin amputation, with full recovery on day 10. The detailed statistical analysis results of this test can be found in Table A2.

### 3.5. Regeneration of the Caudal Fin in Adult Zebrafish

In order to investigate the relationship between zebrafish behavior recovery and caudal fin regeneration, the size of the caudal fin after the amputation was recorded and measured every 5 days for 30 days. From the results, the caudal fin regeneration had started to occur on the earliest observation, which was on 5 dpa, as the whitish stripe of tissue had emerged and enlarged beyond the amputation site (Figure 5A,B). Then, on 10 and 15 dpa, the new tissues persisted at the distal area of outgrowth, while in the proximal part, the white tissues had progressively pigmented and re-differentiated into the mature fin fold (Figure 5C,D). Later, at 20–25 dpa, the size and form of the fin were almost fully restored, leaving a fragile whitish material at the fin margin that may be maintained throughout the entire life of the fish (Figure 5E,F). Finally, based on the literature, the regenerated fin should have regained its original length 30 days after amputation (Figure 5G) [51,65]. Interestingly, based on the graph and previous behavior results, the amputated zebrafish had already been capable of performing a comparably normal behavior at 5 dpa while their caudal fin was only ~20% regenerated (Figure 5H), indicating that the absence of caudal fin did not majorly cause the abnormal behaviors observed on the early observation.

### 3.6. Behavior Recovery after Caudal Fin Amputation by Lidocaine

Several previous studies had demonstrated that the behavioral changes after the fin-amputation procedure are related to the pain elicited from the amputation process. Therefore, it is intriguing to observe the long-term effect and effectiveness of a commonly used drug with pain-relieving properties, lidocaine, in preventing any behavioral responses to the fin amputation during the whole recovery time (10 days) [30,31,45,49,66]. This study is important because, even though lidocaine has previously been demonstrated in reducing the behavioral and physiological symptoms in fin-amputated zebrafish, no study has evaluated its long-term effect on the amputated fish. As we expected, lidocaine appeared to ameliorate the behavioral effects of the fin amputation in that movement complexity was almost fully restored to those seen in control fish, since the overall FD values in this group did not statistically differ from the control group (Figure 6M, Appendix A). Interestingly, this effect lasted during the whole observation period. After further observation, we found that even though behavior recovery was observed in the rescued fish group, some significant differences were shown in some of the behavior endpoints. Regarding the locomotion activity endpoints, while the recovery effect of lidocaine was clearly displayed in the swimming time movement ratio, other endpoints, which are average speed, freezing, and rapid movement time ratios, were still statistically different from the untreated group (Figure 6A–D). These differences were also found in all movement orientation endpoints: average angular velocity and meandering (Figure 6E,F). Furthermore, administration of lidocaine to the fin-amputated fish also induced different fish behaviors as indicated in exploratory behavior endpoints. From these endpoints, the rescued fish group was found to exhibit different preferred positions in the tank during the test than the fin-amputated group and even the untreated group, which was shown by the statistical differences in average distance to the center of the tank, time in the top, middle, and bottom duration of this group (Figure 6G,J–L). Nevertheless, the rescued group still displayed a similar exploratory behavior regarding their transition to the vertical position of the test tank to the control group, which was indicated by the total distance traveled in the top and the number of entries to the top (Figure 6H,I). These differences may affect their movement predictability, resulting in the statistical difference in their entropy values (Figure 6N). In addition, we also provide the behavior results of the non-amputated fish treated with lidocaine. From the results, lidocaine-only treatment did not cause any alterations in their locomotor activity and movement complexity (Figure 6A–D,M). Interestingly, similar to the fin-amputated, lidocaine-treated group, this group also possessed different preferred positions in the tank during the test with a similar vertical exploratory behavior to the control group (Figure 6G–L). Furthermore, slight abnormalities were also found in the movement orientation of this group that also may explain the differences in their entropy level to the untreated group (Figure 6E–F,N). To sum up, the administration of lidocaine to zebrafish after caudal fin amputation helped the fish to maintain their normal behavior performances in the 3D locomotion test during the recovery periods even though their behaviors were still slightly different from those seen in control fish, with some additional minor effects to their behaviors, which also observed after administering lidocaine alone. The detailed statistical analysis results of this test can be found in Table A3.

## 4. Discussions

There are several major findings in the current study. First, we described the impact of amputation of each fin, the dorsal, caudal, anal, pelvic, and pectoral fins, on the behavior performance of adult zebrafish assessed by a 3D locomotion test. Based on the results, even though the amputation of each fin affected the zebrafish behavior, amputation of the caudal fin caused the fish to display the most abnormal behaviors compared to the control group, even at 2 dpa. Next, we investigated the relationship between the behavior recovery of zebrafish after the caudal fin amputation and caudal fin regeneration. To the best of our knowledge, this is the first study to address this issue. Later, it was found that the zebrafish could adapt to the absence of a caudal fin at 5 dpa and performed comparably similar behaviors to the normal fish even though their caudal fin was not fully regenerated. There are several possibilities that might cause this phenomenon. First, the minor regeneration by 5 dpa might already be enough to significantly improve their motion behaviors. Second, the acute loss of the fin may require adaptation of the fish, which might occur as changes in their fine motoric, to compensate for the loss of the main motor when such a high amount of tissue is removed. However, in the present study, we mainly focused on the pain elicited by the fin amputation procedure since we hypothesized that it is the major cause of these observed behavior abnormalities. Therefore, we evaluated the effect of a local anesthetic, lidocaine, on the amputated fish. From the results, lidocaine could ameliorate the behavior performance of fin-amputated fish even though it did not accelerate the process of achieving full behavior recovery. However, lidocaine-treated fish also showed typical behaviors that were not observed in both the control and fin-amputated group but displayed by the fish exposed to lidocaine only, indicating the side effects of this post-amputation treatment protocol, and all of these findings were needed to be further discussed.

As mentioned above, the caudal fin loss resulted in the most severe behavior alteration, especially in their locomotor activity, compared with other fin-amputated groups, which the caudal-fin-amputated fish displayed from 1 hour post-amputation (hpa) to 3 dpa. In line with this finding, similar results had also been described by several previous studies that found a significant reduction in activity levels and movement complexity (reduced fractal dimension), followed by an increment in ventilation after the tail fin amputation in zebrafish. These changes in mechanics are necessary to compensate for the fin loss by growing physical effort [31,67,68]. Furthermore, the amputated fishes preferred to stay at the bottom of the tank and decreased in-tank exploration, as observed in the present study, at least until 3 hpa, while in the first section, this behavior alteration occurred at least until 2 dpa [44,45,49,66]. This dissimilarity might be related to the difference in the basal behaviors of PET and AB zebrafish, as mentioned in our previous study [69]. Nevertheless, these results are highlighting the effects of fin amputation on the exploratory behavior of zebrafish. There are two possible explanations to elucidate this result. First, this phenomenon occurred since the caudal fin is the primary means of transferring momentum from the muscles to the water. Therefore, the absence of this fin caused a substantial reduction in the active area that pushes water and thus, reduced the swimming performance of fish [15]. Second, these behavioral changes may not only be due to a mechanical change in tail length. Instead, they may also come from pain elicited from the fin amputation, since recent studies demonstrated the potential for this procedure to be painful in zebrafish [30,31,45,49,66]. Moreover, we also found that the amputated fishes had already swam quite normally even though their caudal fin had not been fully regenerated. Thus, we verified this argument by conducting a rescue experiment using a pain-relieving drug. In agreement with previous findings, we found that lidocaine administration helped restore the behavior of fin-amputated zebrafish, indicating that the altered behaviors after amputation were also likely caused by the pain [30,31,45,49]. The pain from the amputation procedure is elicited after severance of nerves and blood vessels, which also damages the fish skin and releases an alarm substance that elicits an innate anti-predator response [70]. Therefore, the decrease in fish movements after the amputation is plausible since it indicated that swimming was more painful after the trauma of fin amputation. It also possibly expresses a state of depressive activity guarding behavior following trauma to prevent further damage and pain and promote healing [44,49]. A significant reduction in movement complexity was also demonstrated by fin-amputated individuals in a prior study, together with a decrement in tank exploration that was also found in the present study, which possibly indicates stereotypical behavior [31]. These behavior changes are also found in higher vertebrates, such as farm animals [71]. When they are experiencing pain, these animals become less active and interact less with their environment, making them less conspicuous and able to conserve energy demands if a fight or flight response is required during the presence of a predator. In addition, based on a previous experiment, less movement of the stressed individuals may also be an indicator for fish in avoiding increased aggressive interactions when housed with conspecifics since zebrafish are known to form dominance hierarchies. This hypothesis may also help to explain the higher locomotion activity level exhibited by the amputated fishes when they were individually tested in the 3D locomotion test than when tested in a group in the present study. A prior study demonstrated that, at low stocking density, as in the present study (1 fish/L), the establishment of territoriality and the formation of dominance hierarchies still occurred and were even stronger than high densities, where territories are easier to defend [72]. Generally, this dominant subordinate relationship leads increases aggression between individuals [73,74]. Therefore, the observed low activity in the fishes during the group test might indicate their efforts to avoid fights with other fishes. Furthermore, we also speculated that during the group test, in the tank, the amount of alarm substance, which is made in specialized epidermal club cells and is released when the skin of the fish is damaged, elicited by the group of fin-amputated fish, was higher compared with the condition when the fish were tested individually [75]. A release of this substance had been shown to cause an alarm reaction in neighboring fish, which use their sensitive chemoreceptors to detect the substance, and plays a role in stimulating the behavioral changes in the fishes [44,76]. Therefore, this phenomenon might be absent during the individual test, explaining the behavior differences of the same fishes in both tests. Besides an elicitation of this substance, some studies in fish demonstrated that the response to experiencing stress is also usually indicated by an increment in their rate of respiration and ventilation rate, which is correlated to an increase in plasma cortisol levels, a stress hormone [77]. During a stressful situation in fish, whole-body cortisol is known to increase, and the increment is proportionally depending on the perceived severity of the situation, as demonstrated in a prior study that found the highest levels of cortisol in the fin-amputated fish by using a noninvasive cortisol measurement [66,78]. In addition, since the caudal fin is the most prominent external appendage in zebrafish, an amputation of this fin causes the largest wound area, resulting in the highest level of pain compared with other fins [65].

Previous publications emphasized that the behavioral changes after fin clipping as a response to a noxious event, fin damage, are not transient and last for up to 6 h after the procedure. However, these studies do not specifically test for long-term behavioral effects since the cortisol level returned to baseline levels before 24 h post-stressor [31,48,50]. Interestingly, while another study found that the altered behaviors returned to baseline within 6 h after the procedure and appeared to be normal on the following day, this behavior recovery was not observed in the present study at least until 3 dpa, indicating that the increased anxiety is not a short-lived effect of the procedure [48]. These dissimilarities may be caused by the differences in the housing and behavior test conditions after the fin amputation. In their study, the amputated fish was placed individually in a tank of system water and a novel tank during the recovery period and behavior test, respectively. Meanwhile, in the current study, the fishes were kept in the housing tank and tested in the test tank in a group, which may cause differences in their required recovery time. This assumption was made since several publications highlighted the importance of animal numbers applied in a tank during the fish recovery period. Unfortunately, several conclusions were still contrary to each other. Recent evidence showed that holding zebrafish in groups enhances their recovery from fin amputation [66]. They hypothesized that this phenomenon was caused by social buffering, where social support or being in a familiar group assists could reduce responses to threatening stimuli or events [79,80]. On the other hand, another prior study demonstrated an absence of fish behavior responses recovery when they were housed in pairs during the experiment, which was in line with the current findings. This phenomenon may be explained by the dominance hierarchy, such as social defeat. This system acts as an additional stressor, and thus, recovery could be delayed if a more dominant fish is expending greater energy on maintaining high status or if the paired fish is already experiencing chronic social stress [66]. In addition, as mentioned above, we also speculated that, in the groups of amputated fishes, the amount of alarm substance was high, stimulating the fishes to exhibit innate anti-predator responses [81]. Moreover, a prior study found that lidocaine has less of an impact in reducing the behavioral change in the groups with six fin-amputated fish [44]. This may be due to uptake rates of this small amount of lidocaine (5 ppm); thus, a higher dose may be required when a high number of fishes are applied in the future. Based on the results, future studies are required to verify whether single housing is better than group housing as a method of refinement of the behavior recovery process post fin amputation.

As demonstrated by other publications, we also observed the effectiveness of lidocaine in reducing pain-related responses in zebrafish [30,31,45,49]. Lidocaine, a local anesthetic, is commonly used to manage acute and chronic pain by temporarily inhibiting voltage-gated sodium channels in neuronal plasma membranes and disabling nociceptors in receiving information about painful stimuli [82,83,84]. Interestingly, while lidocaine was already used in medaka with no side effects, our results showed that lidocaine led to slight behavior alterations in zebrafish, especially in terms of their preferred position during the test, to both lidocaine-treated and fin-amputated with lidocaine groups, indicating that the abnormalities were likely caused by lidocaine [85]. This phenomenon may be explained by a well-known anesthesia effect of lidocaine in adult zebrafish, which caused the treated fish seldom to explore the test tank [86,87]. These results were supported by a prior study that administered a higher dose of lidocaine (10 mg/L) via water immersion. From their reports, decreased entries to the top of the tank and locomotor activity were displayed by treated fishes, reflecting the sedative effects of this compound. These effects were also shown in humans, rodents, and zebrafish larvae [87,88,89,90]. However, further clinical and experimental studies to explore the central nervous system (CNS) action of this compound are necessary to elucidate these outcomes.

## 5. Conclusions

Some genotyping studies use fin clip (take usually 10% of the caudal fin tissue) and other studies apply fin amputation (take 30–60% of the caudal fin tissue) [91,92] as their routine procedure to harvest tissue for genomic screening by genotyping mutant and transgenic lines established for experimentation [31,48,93]. This procedure is seen as mild under the Home Office licensing legislation [94]. Interestingly, the results of the present study demonstrated that, by 2 days, zebrafish have not recovered, indicating a prolonged complex change in behavior and the possibility that fin amputation is actually of moderate severity unless analgesia is provided. However, one must keep in mind that, even with treatment with analgesia, specifically lidocaine, side effects could still occur in the treated fish, especially in their behaviors. Nevertheless, since we evaluated swimming behaviors only, it is intriguing to evaluate the potential effect of fin amputation on other behaviors in the future. Observing complex behavior in zebrafish with amputated fins opens new avenues for pain research. In addition, future studies assessing the behavioral effects on zebrafish after a smaller tissue removal than the current study would also be very valuable to provide a new perspective that improves our understanding of the risks of fin clips. To sum up, the present study demonstrates the potential moderate severity of fin amputation in zebrafish and thus, highlights the importance of refinement to current protocols. In addition, when the fin amputation procedure is required in any future research, conditions of the husbandry and usage of analgesia have to be taken into consideration before the experiment to ensure the validity of scientific outputs.

## Figures and Tables

**Figure 1 biology-11-00969-f001:**
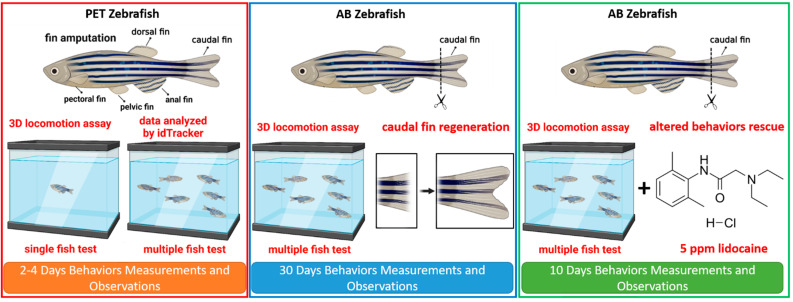
Overview of the experimental design of the current study. In the first section, each fin of the zebrafish was amputated, and the behavior performances of the amputated fishes were assessed in the 3D locomotion test individually and in a group. In the next section, one of the fins that affected the zebrafish behavior performance most was chosen as the fin of interest to evaluate the relation between behavior recovery and fin regeneration in zebrafish. Lastly, we observed the effect of lidocaine to help the fish recover their behaviors after the amputation.

**Figure 2 biology-11-00969-f002:**
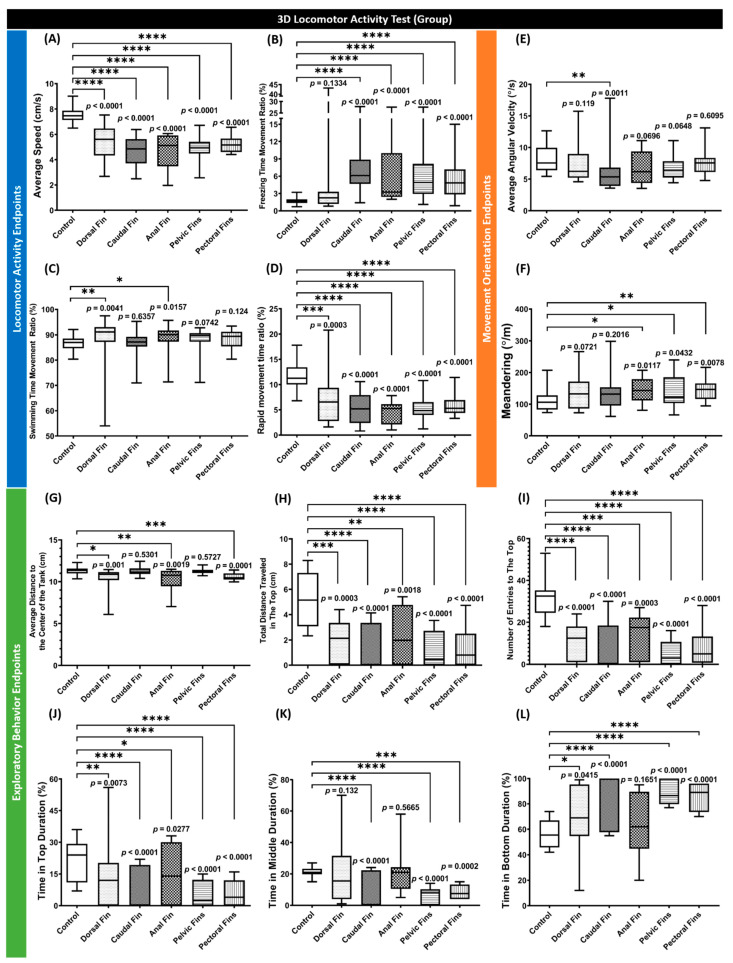
Comparison of grouped zebrafish behaviors in 3D locomotion test among fin-amputated groups and a control group (shoal size = six fishes) at 2 dpa. Twelve endpoints were measured and categorized into three groups. (**A**) Average speed, (**B**) freezing time movement ratio, (**C**) swimming time movement ratio, and (**D**) rapid movement time ratio belongs to the locomotor activity endpoints group, while the movement orientation endpoints group consists of (**E**) average angular velocity and (**F**) meandering. Finally, the exploratory behavior endpoints group is composed of (**G**) average distance to the center of the tank, (**H**) total distance traveled in the top, (**I**) number of entries to the top, (**J**) time in top duration, (**K**) time in middle duration, and (**L**) time in bottom duration. Data are presented as box and whiskers (min to max) and were analyzed by Kruskal–Wallis test continued with uncorrected Dunn’s test (*n* = 18; * *p* < 0.05, ** *p* < 0.01, *** *p* < 0.001, **** *p* < 0.0001).

**Figure 3 biology-11-00969-f003:**
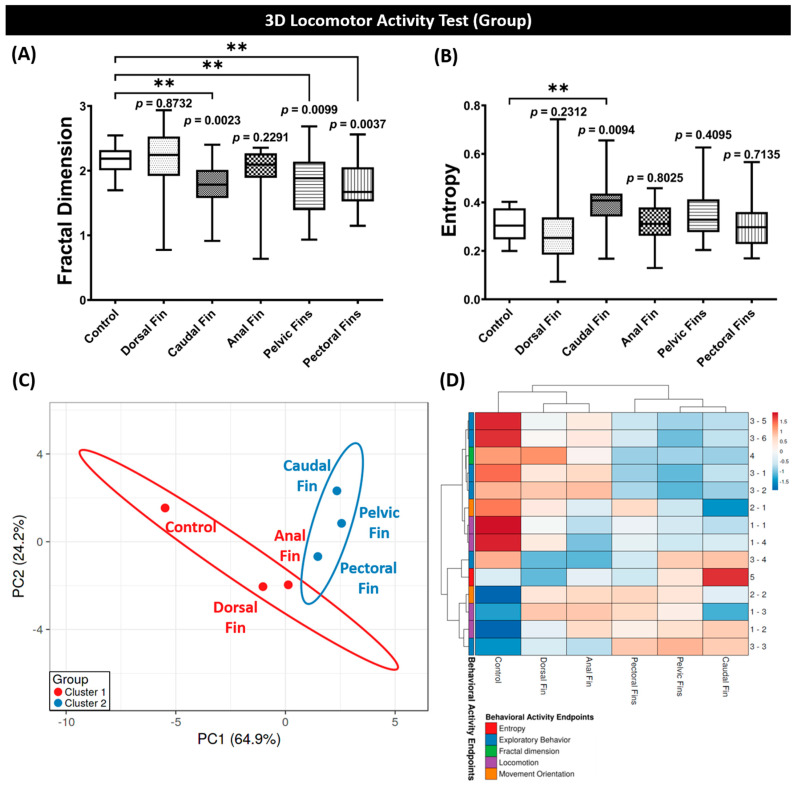
(**A**) Fractal dimension and (**B**) entropy analyses of grouped zebrafish (shoal size = six fishes) swimming patterns in 3D locomotion test at 2 dpa. Data are presented as box and whiskers (min to max) and were analyzed by Kruskal–Wallis test continued with uncorrected Dunn’s test (*n* = 18; ** *p* < 0.01). (**C**) Principal component analysis (PCA) and (**D**) hierarchical clustering analysis of multiple behavior activity endpoints in zebrafish after the fin was amputated. The non-amputated group is included as the control group. In Figure 2C, two major clusters from hierarchical clustering analysis results are marked with red (cluster 1) and blue (cluster 2) colors.

**Figure 4 biology-11-00969-f004:**
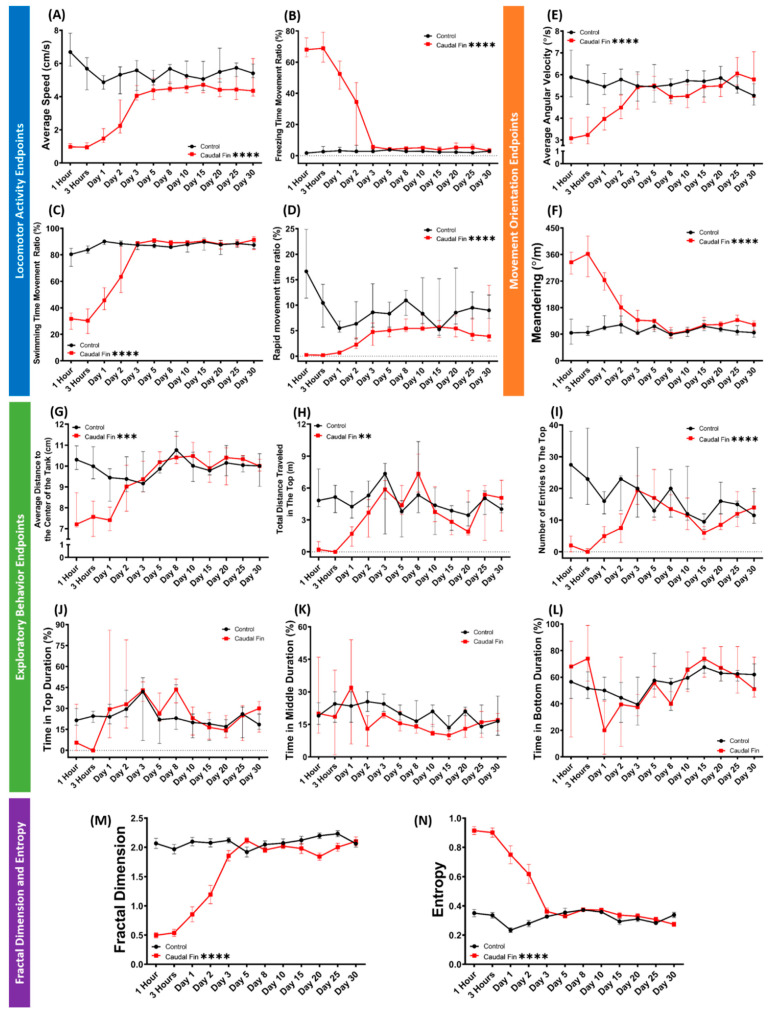
Comparison of grouped zebrafish behaviors in 3D locomotion test between caudal fin-amputated (red) and untreated (black) groups (shoal size = six fishes) during 30 days of observation. Twelve endpoints were measured and categorized into three groups. (**A**) Average speed, (**B**) freezing time movement ratio, (**C**) swimming time movement ratio, and (**D**) rapid movement time ratio belong to the locomotor activity endpoints group, while the movement orientation endpoints group consists of (**E**) average angular velocity and (**F**) meandering. Finally, the exploratory behavior endpoints group is composed of (**G**) average distance to the center of the tank, (**H**) total distance traveled in the top, (**I**) number of entries to the top, (**J**) time in top duration, (**K**) time in middle duration, and (**L**) time in bottom duration. In addition, (**M**) fractal dimension and (**N**) entropy analyses between groups are also added. Data from Figure 4A–L are expressed in the median with 95% CI, while Figure 4M,N are expressed in the mean with SEM. All data were analyzed by mixed models two-way ANOVA continued with Sidak’s multiple comparisons test (*n* = 18, except for caudal-fin-amputated group on day 25–30 (*n* = 17); ** *p* < 0.01, *** *p* < 0.001, **** *p* < 0.0001).

**Figure 5 biology-11-00969-f005:**
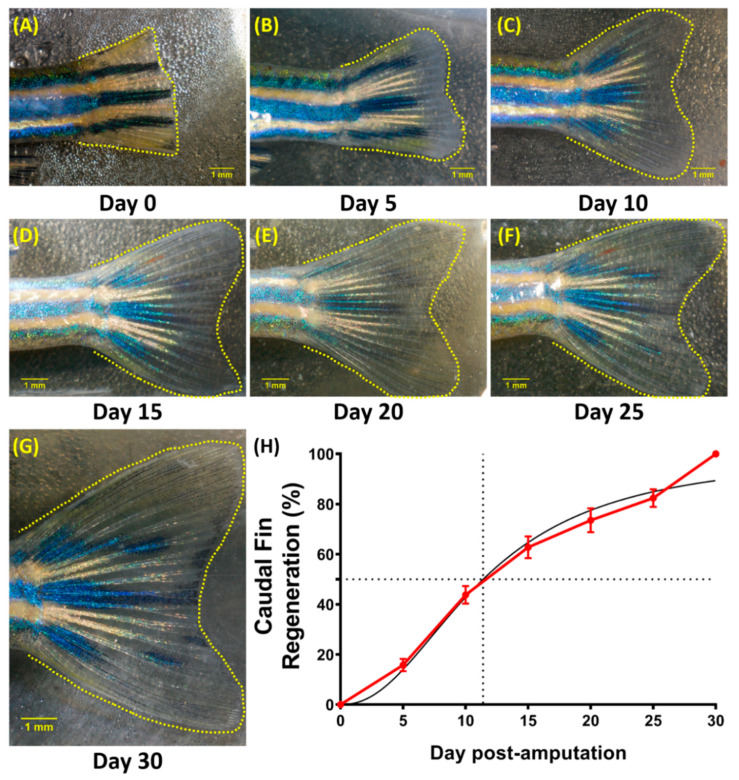
Representative images of a caudal fin regeneration from a single fish at (**A**) 0, (**B**) 5, (**C**) 10, (**D**) 15 (**E**) 20 (**F**) 25, and (**G**) 30 days post-amputation. (**H**) Caudal fin regeneration curve during 30-day observation (red line) with the mathematical model of regeneration percentage prediction (normalized) (black line), which indicates that 50% of regeneration occurred approximately 11 days after the amputation. The data are expressed as the mean with SEM (*n* = 18).

**Figure 6 biology-11-00969-f006:**
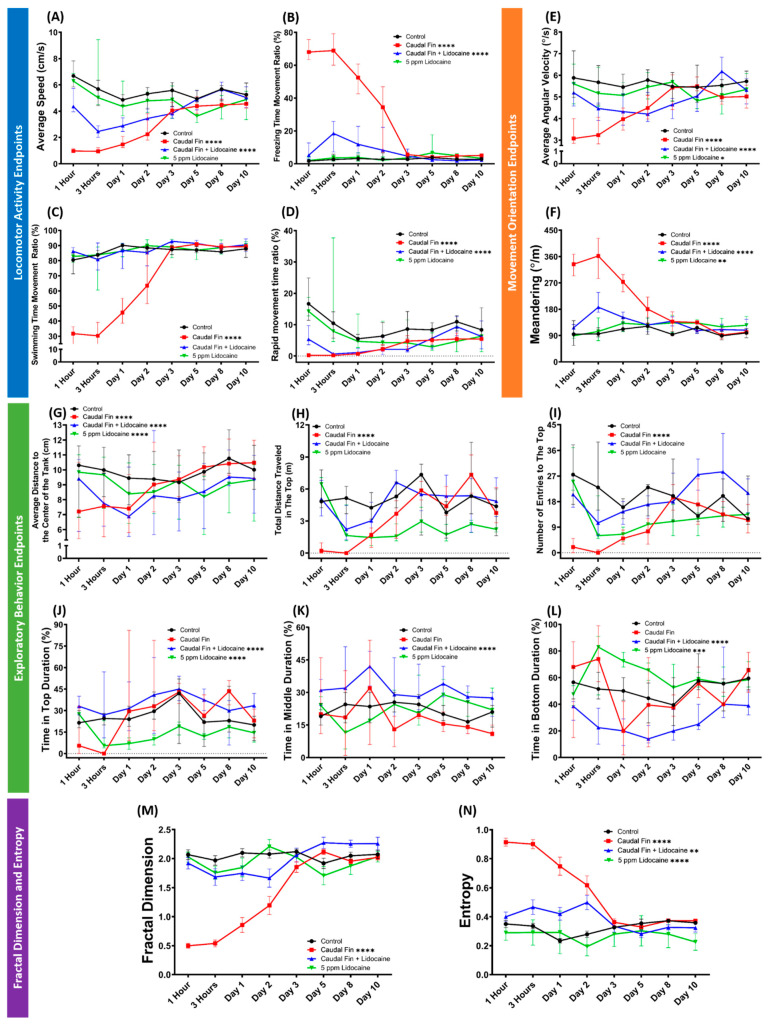
Comparison of grouped zebrafish behaviors in 3D locomotion test between caudal-fin-amputated group (red), caudal-fin-amputated + 5 ppm lidocaine group (blue), 5 ppm lidocaine-treated group (green), and untreated (black) group (shoal size = six fishes) over 10 days observation. Twelve endpoints were measured and categorized into three groups. (**A**) Average speed, (**B**) freezing time movement ratio, (**C**) swimming time movement ratio, and (**D**) rapid movement time ratio belong to the locomotor activity endpoints group, while the movement orientation endpoints group consists of (**E**) average angular velocity and (**F**) meandering. Finally, the exploratory behavior endpoints group is composed of (**G**) average distance to the center of the tank, (**H**) total distance traveled in the top, (**I**) number of entries to the top, (**J**) time in top duration, (**K**) time in middle duration, and (**L**) time in bottom duration. In addition, (**M**) fractal dimension and (**N**) entropy analyses between groups are also added. Data from Figure 6A–L are expressed in the median with 95% CI, while Figure 6M,N are expressed in the mean with SEM. All data were analyzed by mixed models two-way ANOVA continued with Sidak’s multiple comparisons test (*n* = 18, except for the caudal-fin-amputated + lidocaine group on day 2–3 (*n* =17) and 5–10 (*n* = 16) and lidocaine-treated group on day 5 (*n* =17) and 8–10 (*n* = 16); ** *p* < 0.01, *** *p* < 0.001, **** *p* < 0.0001).

## Data Availability

The datasets used and/or analyzed during the current study are available from the corresponding author on reasonable request.

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
