# Peer review of "Acute and Chronic Effects of Fin Amputation on Behavior Performance of Adult Zebrafish in 3D Locomotion Test Assessed with Fractal Dimension and Entropy Analyses and Their Relationship to Fin Regeneration"

_biology, 2022, doi:10.3390/biology11070969_

Round 1

Reviewer 1 Report

The presented paper about acute and chronic effects of fin amputation on behaviour in zebrafish by Audira et al. targets a very important topic especially with regard to animal welfare in research. The applied methods and evaluation endpoints are highly sophisticated and reveal a detailed insight in behavioural effects. This will lead the field forward in evaluating animal welfare.

Still, there are some points that need clarification first. One main point is the size of the amputation and the conclusions drawn from that. In this paper as well as in most other papers that the authors are citing the so-called “fin clip” involves more that 40% of the caudal fin tissue, in this study it is close to a full amputation. While this is a size of tissue removed that is standard for regeneration studies, many people and also the authors of this paper refer to the fin clip for genotyping as a comparable procedure. This is not the case. Most labs only use 5-10% of the caudal fin tissue for genotyping. This is highly recommended with regard to the 3R standards that should be applied to animal studies. And this should also be discussed more in detail in this paper (see also comments below). Especially, the caudal fin is the main motor of locomotion as it is the one mostly supported by muscular movement (although not containing muscle fibres itself). When removing as much tissue as it is described in this study (and the others) it is very probable that a fish will show severe struggling to the new “feeling” when such an important part of the body is changed in such an elementary way. The typical fin clip for genotyping probably will have much fewer effects. Thus, I would highly recommend that additional studies are performed that include fin clips of different sizes to compare if the authors want to draw conclusions for the fin clip for genotyping, but it would be of high relevance then.

Detailed comments:

Major points:

Line 196: it is mentioned that fish receiving a fin amputation also receive amoxicillin, but not the other treatment or control groups. Are there any tests that amoxicillin has no effects on the studies?

Line 211, Line 444-445, line 619: It would be very important to differentiate more precisely between fin clip as it is routinely done for genotyping taking usually only maybe 10% of the caudal fin tissue (if not even less) and fin amputation as it is used here as well as in most papers studying effects of “fin clip“ using mostly 40% or even more tissue as in this study.

Line 223: why are the tests done in a water temperature significantly lower than the husbandry temperature as this may influence results? Wound might be more sensitive to temperature effect that intact skin.

Line 251: The authors describe an imaging of fin regeneration in amputated fish. How was this done? Where the fish be placed in MS-222 or anything else to take the pictures? This may influence the results compared to the other groups if these weren’t anesthetized.

Line 345 as well as 557-/560 and 592/593: Would that lead to the suggestion to use single housing as a method of refinement when doing fin clips?

Line 400-402 and figure 4J-L: How is the difference compared to figure 2 J-L explained where amputation caused a change in the preference of specific compartments in the tank?

Figure 4 A-c: Especially day 3 has a very high standard deviation. How is this explained?

Line 510: The authors conclude that pain must be part of the behavioural changes as the behaviour is already recovered while the regeneration is still quite incomplete. While this is a possible explanation two other explanations should be considered as well: 1st: the minor regeneration by day 5 might already be enough to significantly improve motion behaviour. 2nd: The acute loss of the whole fin may require adaptation of the fish to compensate for the loss of the main motor and the changed fine motoric that occurs when such a high amount of tissue is removed (see also line 535, 544-546, ).

Line 519: The authors describe that the main changes occur in locomotor activity. I would expect that since this is the main "motor" for propulsion and the muscular strength involved in the movement. The other fins are not as well supported by muscles.

Lines 523, 538, 543, 546, 558, 566, 586, 598: All the studies cited are describing a fin amputation of 40% (or even more tissue) That should be emphasized more in detail.

Line 609: The authors refer to lidocaine as the substance inducing behavioural changes. Additional effects may also come from tricaine during the amputation procedure as there is no tricaine in the control group and there is no negative control for tricaine.

Minor points:

Line 175:  The description „fish were fed twice with … brine shrimp or … dry food.“ Is not very precise. Was it brine-shrimp once a day and dry food once a day or one day brine shrimp twice and the other day dry food twice? Or irregular composition of food?

Line 193: the concentration of MS-222 is missing in the methods

Line 391: I found the reference of video A1 is misleading here in the way it is presented. It sounds like there is a video of the regeneration of the fin, not of behavioural videos. That should be made clearer.

Author Response

Comments and Suggestions for Authors

The presented paper about acute and chronic effects of fin amputation on behaviour in zebrafish by Audira et al. targets a very important topic especially with regard to animal welfare in research. The applied methods and evaluation endpoints are highly sophisticated and reveal a detailed insight in behavioural effects. This will lead the field forward in evaluating animal welfare. Still, there are some points that need clarification first. One main point is the size of the amputation and the conclusions drawn from that. In this paper as well as in most other papers that the authors are citing the so-called “fin clip” involves more that 40% of the caudal fin tissue, in this study it is close to a full amputation. While this is a size of tissue removed that is standard for regeneration studies, many people and also the authors of this paper refer to the fin clip for genotyping as a comparable procedure. This is not the case. Most labs only use 5-10% of the caudal fin tissue for genotyping. This is highly recommended with regard to the 3R standards that should be applied to animal studies. And this should also be discussed more in detail in this paper (see also comments below). Especially, the caudal fin is the main motor of locomotion as it is the one mostly supported by muscular movement (although not containing muscle fibres itself). When removing as much tissue as it is described in this study (and the others) it is very probable that a fish will show severe struggling to the new “feeling” when such an important part of the body is changed in such an elementary way. The typical fin clip for genotyping probably will have much fewer effects. Thus, I would highly recommend that additional studies are performed that include fin clips of different sizes to compare if the authors want to draw conclusions for the fin clip for genotyping, but it would be of high relevance then.

Detailed comments:

Major points:

Line 196: it is mentioned that fish receiving a fin amputation also receive amoxicillin, but not the other treatment or control groups. Are there any tests that amoxicillin has no effects on the studies?

Thank you for the question. To the best of the authors’ knowledge, only a few studies evaluate the effects of amoxicillin on zebrafish, especially their behaviours. Up to now, the authors can find only one study that might relate to this issue. However, even though they found that amoxicillin could change zebrafish behaviour, this finding does not conclude that the used amoxicillin in the present study has similar effects since, in their study, the concentration of amoxicillin was higher (100 ppm) than in the current study (0.5 ppm). Moreover, in their study, zebrafish were exposed to this high concentration of amoxicillin for 7 days while here, a single drop of amoxicillin was given only one time after the amputation procedure and only at the amputation site. Therefore, the authors believed that the use of amoxicillin in the present study was very minor and likely would not cause any significant differences in terms of the behaviours between the control and amputated groups. Furthermore, although it caused some effects, this step still belongs to the fin amputation procedure that is generally applied in various researches, which is included in the main aim of the present study. Therefore, whether this antibiotic could affect their behaviours or not, the current findings still highlight the effects of the whole fin amputation procedure on zebrafish behaviours. In addition, to provide more clarity regarding the used amoxicillin, additional information was added to the manuscript.

Gonçalves, Cinara L., et al. "Exposure to a high dose of amoxicillin causes behavioral changes and oxidative stress in young zebrafish." Metabolic Brain Disease 35.8 (2020): 1407-1416.

Line 211, Line 444-445, line 619: It would be very important to differentiate more precisely between fin clip as it is routinely done for genotyping taking usually only maybe 10% of the caudal fin tissue (if not even less) and fin amputation as it is used here as well as in most papers studying effects of “fin clip“ using mostly 40% or even more tissue as in this study.

The authors appreciated the reviewer for raising this important point; thus, all of the unsuitable ‘fin-clip’ terms were changed to ‘fin-amputation’ terms to differentiate these terms more precisely as the reviewer suggested. In addition, additional information was added to the conclusions section to enhance the importance of the current finding, especially in genotyping procedure, since although a routinely genotyping procedure in some laboratories only takes 10% of the caudal fin tissue, as the reviewer mentioned, there may also other facilities that take more than 10% of the tissue for genotyping procedure. This assumption was taken since there were some genotyping protocols that instruct to cut either 2-3 mm (30-60%) of caudal fin in order for zebrafish to swim normally) or no more than 50% of the fin area. Therefore, this important information was added to the manuscript to avoid confusion between these two terms and highlight the significance of the current study.

Xing, L.; Quist, T.S.; Stevenson, T.J.; Dahlem, T.J.; Bonkowsky, J.L. Rapid and efficient zebrafish genotyping using PCR with high-resolution melt analysis. JoVE (Journal of Visualized Experiments) 2014, e51138.

UCSF Institutional Animal Care and Use Program. Fin Clipping of Zebrafish. Available online: https://iacuc.ucsf.edu/sites/g/files/tkssra751/f/wysiwyg/STD%20PROCEDURE%20-%20Aquatic%20-%20Fin%20Clipping%20of%20Zebrafish.pdf.

Line 223: why are the tests done in a water temperature significantly lower than the husbandry temperature as this may influence results? Wound might be more sensitive to temperature effect that intact skin.

Thank you for the question. Actually, there was a mistake regarding the information on the locomotion test condition. Here, the behaviour tests used water with the same temperature as the husbandry temperature as in fact, the water was taken from the husbandry facility, and this test was conducted in a temperature-controlled room. This information in the manuscript had been corrected accordingly.

Line 251: The authors describe an imaging of fin regeneration in amputated fish. How was this done? Where the fish be placed in MS-222 or anything else to take the pictures? This may influence the results compared to the other groups if these weren’t anesthetized.

The authors appreciated the reviewer for the suggestions. It is true that the information regarding the imaging process of fin regeneration in amputated fish was not described clearly in the previous version of the manuscript. Therefore, in this updated version, some information regarding this process was added, including the used anesthetize agent, the apparatus to maintain the fish position, and the recovery process, as the reviewer’s suggestion.

Line 345 as well as 557-/560 and 592/593: Would that lead to the suggestion to use single housing as a method of refinement when doing fin clips?

Thank you for the question. Actually, the group housing was chosen in the present study since there are still contradiction between several prior studies regarding these two refinement methods as mentioned in lines 626-627. However, based on the current findings, as the reviewer mentioned, there is a possibility that single housing is better than group housing as a method of refinement after the fin amputation procedure since here; the treated fishes displayed a relatively slow behaviour recovery after fin amputation. This possibility is interesting to be verified in future studies. Nevertheless, the results of the group housing method described in the present study might help in elucidating the outcome differences between single and group housing methods. In addition, this input was added to the discussion part, mentioning the possibility of a better output of single housing than group housing methods.

Line 400-402 and figure 4J-L: How is the difference compared to figure 2 J-L explained where amputation caused a change in the preference of specific compartments in the tank?

The authors thanked the reviewer for the detailed questions. As the reviewer mentioned, indeed, there is a difference in the preference of fishes in the specific compartments of the tank between Figure 2 and Figure 4 although their caudal fin was amputated. This dissimilarity might be caused by the differences in the basal behaviours between PET and AB zebrafish as mentioned in the author’s prior study. Nevertheless, these differences are still indicating the overall behaviour alterations, especially in their exploratory behaviours, caused by the fin amputation. This important information was included in the discussion part to avoid confusion, according to the reviewer’s suggestion.

Audira, Gilbert, et al. "Which zebrafish strains are more suitable to perform behavioral studies? A comprehensive comparison by phenomic approach." Biology 9.8 (2020): 200.

Figure 4 A-c: Especially day 3 has a very high standard deviation. How is this explained?

Thank you for the question. The authors suspected that the very high standard deviation observed on day 3 might be related to the unstable behaviours of the tested fishes since by this time point; they had already been subjected to the test consecutively for several days. Therefore, these continued behaviour tests might result in mild exhaustion in the fish and thus, slightly affect their behaviour performances during the test.  

Line 510: The authors conclude that pain must be part of the behavioural changes as the behaviour is already recovered while the regeneration is still quite incomplete. While this is a possible explanation two other explanations should be considered as well: 1st: the minor regeneration by day 5 might already be enough to significantly improve motion behaviour. 2nd: The acute loss of the whole fin may require adaptation of the fish to compensate for the loss of the main motor and the changed fine motoric that occurs when such a high amount of tissue is removed (see also line 535, 544-546, ).

The authors appreciated the reviewer for the constructive suggestions. As the reviewer stated, it is true that there are several possible explanations that have to be considered as part of the observed behaviour recovery while the regeneration is still quite incomplete. Therefore, these possibilities were included in the manuscript, specifically in the discussion part, as the reviewer suggested, without neglecting the fact that the present study was mainly focused on the pain elicited by the fin amputation procedure as the major cause of the observed behaviour abnormalities.

Line 519: The authors describe that the main changes occur in locomotor activity. I would expect that since this is the main "motor" for propulsion and the muscular strength involved in the movement. The other fins are not as well supported by muscles.

The authors understood the reviewer’s point of view. The caudal fin indeed plays a role in propulsion and the muscular strength involved in the movement, as the reviewer stated. However, based on the literature studies, the pectoral fins are the fin that acts as primary propulsors during rhythmic swimming and in arrhythmic movements such as braking and maneuverings, and the caudal fin assists this fin to achieve higher relative speeds. Therefore, initially, the authors also expected that amputation in the pectoral fins resulted in the most prominent changes in their locomotion while in the caudal fin amputated group, the pectoral fin might compensate for the absence of the caudal fin in their locomotor activity performance. In addition, this line is also intended to emphasize that the most prominent alteration was observed in their locomotion although other abnormalities were also found in other behaviour endpoints.

Lines 523, 538, 543, 546, 558, 566, 586, 598: All the studies cited are describing a fin amputation of 40% (or even more tissue) That should be emphasized more in detail.

Thank you for the detailed corrections. The term ‘fin-clip’ used in those lines was revised to ‘fin amputation’ as the reviewer suggested since all of those studies performed a 40% fin amputation although in those reports they use ‘fin-clip’ to describe their procedure.

Line 609: The authors refer to lidocaine as the substance inducing behavioural changes. Additional effects may also come from tricaine during the amputation procedure as there is no tricaine in the control group and there is no negative control for tricaine.

The authors thanked the reviewer for the comment. Actually, a prior study demonstrated that tricaine has no effect on several commonly used behavioural parameters and the 30-min postponement after anesthesia may be unnecessary before conducting behavioural observations. Based on this finding, negative control for tricaine was not provided and this consideration was also taken to reduce the animal usage in the current study.

Nordgreen, Janicke, et al. "Behavioural effects of the commonly used fish anaesthetic tricaine methanesulfonate (MS-222) on zebrafish (Danio rerio) and its relevance for the acetic acid pain test." PLoS One 9.3 (2014): e92116.

Minor points:

Line 175:  The description „fish were fed twice with … brine shrimp or … dry food.“ Is not very precise. Was it brine-shrimp once a day and dry food once a day or one day brine shrimp twice and the other day dry food twice? Or irregular composition of food?

Thank you for the questions. The authors were aware that there was a mistake in describing some of the fish husbandry procedures, which lead to confusion. Therefore, the information was revised to avoid misinformation accordingly.

Line 193: the concentration of MS-222 is missing in the methods

The authors appreciated the reviewer for the detailed correction. Here, MS-222 in 0.1% concentration was used. This information was added to the manuscript, specifically in the materials and methods section.

Line 391: I found the reference of video A1 is misleading here in the way it is presented. It sounds like there is a video of the regeneration of the fin, not of behavioural videos. That should be made clearer.

Thank you for raising this point. However, the authors believe that the reference of video A1 is already placed in the correct section since it is mentioned in the section that specifically discusses the results of the behaviour recovery of the fin-amputated fishes while the discussion of the fin regeneration was written in another section of the results part.

Reviewer 2 Report

Comments Manuscript biology-1747557-peer-review-v1 by Audira and colleagues, entitled ‘Acute and Chronic Effects of Fin Amputation on Behavior Performance of Adult Zebrafish in 3D Locomotion Test Assessed with Fractal Dimension and Entropy Analyses and Their Relationship to Fin Regeneration’.

Audira and colleagues have performed a relevant study examining the impact of fin amputation on locomotory activities in zebrafish (Danio rerio). Results showed that caudal-fin amputation has the most pronounced locomotory activities’ alteration compared to the other fin-amputations. More interesting is the possibility of considering several behavioral indices on locomotory activities to assess the welfare of fish, which allows authors to make robust conclusions about the impact of fin-amputation on fish mobility. As a secondary test, the authors revealed differences in the behavior recovery process in fish treated with lidocaine, emphasizing the importance of analgesia usage in fin amputation.

The study is well-executed, and it was a great read. I have some comments regarding the analytical methods and their presentation, which the authors should consider to hopefully improve the quality and readability of their manuscript.

Line 165: specify fish life-stage

Lines 183-185: I would recommend changing this sentence. In the present form, the soundness sounds distorted for animal welfare, because researchers can use as many fish as want without considering legislation on conducting animal experiments. The authors should rephrase the sentence by accentuating the limited number of AB strains rather than the infinite availability of WT from the local shop.

Lines 199-202: Some technical information is missing, such as the value of effect size (based on previous research or hypothetical strength), and the type of statistical test.

Lines 229-232:  Details of video recording are missing: i) the distance of the camera from the tank; ii) the number of frames x second. According to the last point, this information is important to understand the number of X, Y, and Z coordinates (each frame, I guess) used for the calculation of di points. Although this information is reported in a previous study (https://doi.org/10.3390/inventions3010011), the authors should report it for clarifying how they perform the test.

Statistical analysis.

Authors have analyzed different behavioral indexes by considering single and group fish. Besides a mixed-model is necessary for analyzing group data (random factor: group), I can not understand how mixed-model has been used for individual data when authors considered the overall/mean measured index and not the time-series sequence. I suggested providing more information on how authors analyzed such behavioral indices.

I would suggest authors add more detail in the section ‘Principal components analysis (PCA) and clustering analyses. A brief description of the single value decomposition and a relative reference of previous studies that applied it for analyzing similar data is necessary. Second, the authors reported a hierarchical clustering analysis without mentioning how they have done it (i.e., the metric or the measures of similarity, linkage criteria).

Because of the non-normally distributed data, bar plots should be replaced with boxplots.

Finally, I would ask the authors to discuss the observed difference in locomotory behavior when fish are isolated or group. 

Author Response

Comments and Suggestions for Authors

Comments Manuscript biology-1747557-peer-review-v1 by Audira and colleagues, entitled ‘Acute and Chronic Effects of Fin Amputation on Behavior Performance of Adult Zebrafish in 3D Locomotion Test Assessed with Fractal Dimension and Entropy Analyses and Their Relationship to Fin Regeneration’.

Audira and colleagues have performed a relevant study examining the impact of fin amputation on locomotory activities in zebrafish (Danio rerio). Results showed that caudal-fin amputation has the most pronounced locomotory activities’ alteration compared to the other fin-amputations. More interesting is the possibility of considering several behavioral indices on locomotory activities to assess the welfare of fish, which allows authors to make robust conclusions about the impact of fin-amputation on fish mobility. As a secondary test, the authors revealed differences in the behavior recovery process in fish treated with lidocaine, emphasizing the importance of analgesia usage in fin amputation.

The study is well-executed, and it was a great read. I have some comments regarding the analytical methods and their presentation, which the authors should consider to hopefully improve the quality and readability of their manuscript.

Line 165: specify fish life-stage

Thank you for the correction. Here, adult zebrafish, which ranged from 4 to 6 months old were used in the experiment. This crucial information was added to the manuscript, specifically in the materials and methods section according to the reviewer’s suggestion.

Lines 183-185: I would recommend changing this sentence. In the present form, the soundness sounds distorted for animal welfare, because researchers can use as many fish as want without considering legislation on conducting animal experiments. The authors should rephrase the sentence by accentuating the limited number of AB strains rather than the infinite availability of WT from the local shop.

The authors strongly agreed with the correction. It is true that in the previous version, the sentence may cause misleading to the readers regarding the usage of PET zebrafish which might lead to overexploitation of this zebrafish strain in the future. Therefore, the mentioned sentence was rephrased to emphasize the limited number of AB strains than the infinite availability of PET strains from the local shop as the reviewer suggested.

Lines 199-202: Some technical information is missing, such as the value of effect size (based on previous research or hypothetical strength), and the type of statistical test.

Thank you for the reminder. More information regarding the used statistical test to determine the sample size that was applied in the present study was added to this section. In addition, the average value of effect size calculation results for both group and individual data was also added to the manuscript. Here, Cohen’s d method was used to calculate the effect size between untreated and every fin amputated group and found that the values were 3.23 and 0.63 for group and individual data, respectively, indicating that the current finding has practical significance.

Lines 229-232:  Details of video recording are missing: i) the distance of the camera from the tank; ii) the number of frames x second. According to the last point, this information is important to understand the number of X, Y, and Z coordinates (each frame, I guess) used for the calculation of di points. Although this information is reported in a previous study (https://doi.org/10.3390/inventions3010011), the authors should report it for clarifying how they perform the test.

The authors appreciated the reviewer for the constructive suggestions. As the reviewer mentioned, although the details of video recording are reported in the previous study, the authors also agreed that this information is still required to be reported in the manuscript to clarify how the tests were performed. Therefore, some important information regarding the behaviour tests was added to the manuscript, including the distance of the camera from the test tank and the video quality information (resolution and fps), according to the reviewer’s suggestion.

Statistical analysis.

Authors have analyzed different behavioral indexes by considering single and group fish. Besides a mixed-model is necessary for analyzing group data (random factor: group), I can not understand how mixed-model has been used for individual data when authors considered the overall/mean measured index and not the time-series sequence. I suggested providing more information on how authors analyzed such behavioral indices.

Thank you for the comment. Actually, for the data of individual and grouped zebrafish behaviour in the first section of the study (Figures 2, 3, and A1), the authors did not use a mixed-model analysis for the statistical test since generally, the mixed-effects model approach available in the used statistic software (GraphPad Prism) is used to analyze repeated measures data. In the current study, mixed-effects analysis was applied to all of the data in the second and last sections of the experiment (Figures 4 and 6) since in those data, there are two factors that may affect the behaviours of the fish, which were amputation treatment and time. For this mixed-effects model approach, two-way ANOVA with Geisser-Greenhouse’s correction continued with Sidak’s multiple comparisons test was chosen. While for the data from the first section, Kruskal-Wallis followed with Dunn’s multiple comparisons test was used since the data were not a repeated measures data (individual data). To avoid confusion, more information was added to the manuscript, especially in the materials and methods section.

I would suggest authors add more detail in the section ‘Principal components analysis (PCA) and clustering analyses. A brief description of the single value decomposition and a relative reference of previous studies that applied it for analyzing similar data is necessary. Second, the authors reported a hierarchical clustering analysis without mentioning how they have done it (i.e., the metric or the measures of similarity, linkage criteria).

The authors appreciated the suggestion and also felt the necessity of a brief description in the section on PCA and clustering analysis to provide more clarity on the used methods and help the readers that are not familiar with these approaches in understanding the results. Therefore, a brief description regarding the SVD with the imputation method used for the PCA was added to the section. Next, the authors also agreed that there was a minimum description regarding the used heatmap and hierarchical clustering method available in the previous version. Thus, several important pieces of information regarding this step were also included in the manuscript, including the clustering process in general, the clustering distance method, and the linkage method applied in this method. In addition, several references that applied this method for analyzing similar data were also added according to the suggestions from the reviewer.

Because of the non-normally distributed data, bar plots should be replaced with boxplots.

Thank you for the suggestion. The authors agreed with the reviewer that the non-normally distributed data are more suitable if presented as boxplots. Therefore, the entire bar plots in the manuscript were changed to box and whiskers (min to max) as the reviewer’s suggestion.

Finally, I would ask the authors to discuss the observed difference in locomotory behavior when fish are isolated or group. 

The authors thanked the reviewer for the constructive comment. The authors agreed that in the previous version, the discussions of the behaviour performance differences between group and individual fish were not sufficient enough. Therefore, in this version, some information was added to the discussion part to help in explaining the occurrence of these behaviour differences. Here, the authors provide two possibilities that might cause this phenomenon. First, the differences might be related to their responses toward a dominant subordinate relationship that may occur during the group test, which can lead to increased aggression between individuals. Second, this result occurred might be due to the high amount of alarm substance that was released when the skin of the fish is damaged compared to the condition during the individual test. The substance had been demonstrated to cause an alarm reaction in neighboring fish and thus, plays a role in stimulating their behavioural changes.

Round 2

Reviewer 1 Report

The manuscript was improved sigificantly. I would like to thank the authors for their work. Please keep in mind that a comparison between the results descriped in this paper and a tissue removal of a much smaller size to see whether there are different results would be very valuable.

There is only one small typo in line 605 (starting a new sentence with a big letter ("Some").

Author Response

The manuscript was improved significantly. I would like to thank the authors for their work. Please keep in mind that a comparison between the results descriped in this paper and a tissue removal of a much smaller size to see whether there are different results would be very valuable.

The authors also would like to thank the reviewer(s) for their constructive and detailed comments and corrections. Finally, in this revised version, the authors also added some information in the conclusion part that highlights the importance of future studies evaluating the effects, especially in the behaviors, of a smaller tissue removal on zebrafish to complement the results from the present study according to the reviewer’s comment. The authors hoped that the combination of these studies could provide a new perspective that improves our understanding of the risks of fin clips.

There is only one small typo in line 605 (starting a new sentence with a big letter ("Some").

Thank you for the detailed correction. The mistyping was corrected according to the reviewer’s suggestion.